# Sea-Cucumber-like Microstructure Polyoxometalate/TiO_2_ Nanocomposite Electrode for High-Performance Electrochromic Energy Storage Devices

**DOI:** 10.3390/molecules28062634

**Published:** 2023-03-14

**Authors:** Xiaoshu Qu, Lili Zhou, Zefeng Liu, Zeyu Wang, Jilong Wang, Xiaoyang Yu, Hua Jin, Yanyan Yang

**Affiliations:** College of Chemical and Pharmaceutical Engineering, Jilin Institute of Chemical Technology, Jilin 132022, China

**Keywords:** electrochromism, energy storage, polyoxometalate, nanostructures, EESD

## Abstract

The key challenge in the practical application of electrochromic energy storage devices (EESDs) is the fabrication of high-performance electrode materials. Herein, we deposited K_7_[La(H_2_O)_x_(α_2_-P_2_W_17_O_61_)] (P_2_W_17_La) onto TiO_2_ nanowires (NW) to construct an NW–P_2_W_17_La nanocomposite using a layer-by-layer self-assembly method. In contrast to the pure P_2_W_17_La films, the nanocomposite exhibits enhanced electrochromic and electrochemical performance owing to the 3D sea-cucumber-like microstructure. An EESD using the NW–P_2_W_17_La film as the cathode exhibited outstanding electrochromic and energy storage properties, with high optical modulation (48.6% at 605 nm), high switching speeds (t_coloring_ = 15 s, t_bleaching_ = 4 s), and high area capacitance (5.72 mF cm^−2^ at 0.15 mA cm^−2^). The device can reversibly switch between transparent and dark blue during the charge/discharge process, indicating that electrochromic contrast can be used as a quantitative indicator of the energy storage status.

## 1. Introduction

Amidst increasing environmental pollution and the gradual exhaustion of non-renewable energy resources, the development of controllable devices for energy storage, conversion, and protection, such as smart windows [1,2,3], micro-supercapacitors [4,5], lithium-ion batteries [6,7], electrochemical capacitors [8,9], and solar cells [10,11], has become a global priority. Electrochromic (EC) devices and batteries/pseudocapacitors have similar electrode materials, device structures, and faradaic reactions [12]. Integrating electrochromic and energy storage properties into a single platform to construct electrochromic energy storage devices (EESDs) enables versatile use of energy and significantly reduces the cost of energy storage [13,14,15,16]. EESDs change color in real time based on the level of stored energy and have the potential to be used in new smart electronic products [17], smart clothing [18], and military equipment [19]. 

A challenge in the development of EESDs is the fabrication of high-performance electrode materials. Recently, bifunctional electrode materials, such as WO_3_ [20], V_2_O_5_ [21], NiO [22], and Prussian white [23], have attracted significant attention. Deng et al. [24] fabricated MnO_2_/WO_3_ EC electrodes with ultra-long stability and excellent optical modulation, and assembled all-solid-state zinc-ion supercapacitors with high coloration efficiency and cycling stability. Zhang et al. [25] constructed asymmetric electrochromic supercapacitors based on a new conjugated polymer (PETC) and V_2_O_5_ ion-storage material, which outperformed other conjugated polymer-based electrochromic supercapacitors in terms of electrochemical performance. Polyoxometalates (POMs) are ideal EC and electrochemical storage materials. Owing to their unique multi-electron storage capacity and rich redox properties, POMs have been extensively studied for use in EC and energy-related technologies [26,27,28,29]. Fan et al. [30] prepared a self-supporting composite based on H_3_PMo_12_O_40_ heterogeneous blue-modified reduced graphite oxide (PAHB/RGO), which had a specific capacitance of 395 F g^−1^ at 1 A g^−1^. The symmetrical solid-state supercapacitor assembled with PAHB/RGO electrodes had excellent specific energy and power. Zhang et al. [31] prepared flexible amorphous isopolytungstate electrochromic devices using inkjet printing technology and observed quantitative color changes in response to different voltages.

There is currently a dearth of research on POM-based EESDs. Recently, our research group investigated several high-performance electrochromic energy storage bifunctional composites containing saturated [32], lacunary [33], and substituted-type [34] POMs. In this study, we further developed a POM-based EESD and studied its EC and energy storage performance. We deposited K_7_[La(H_2_O)_x_(α_2_-P_2_W_17_O_61_)] (abbreviated as P_2_W_17_La) nanoparticles onto TiO_2_ nanowires (TiO_2_ NW) to obtain an electrochromic energy storage bifunctional nanocomposite (NW–P_2_W_17_La) via a combined hydrothermal and layer-by-layer (LbL) self-assembly method. The 3D sea-cucumber-like microstructure of NW–P_2_W_17_La enhanced the coverage of the surface, exposed more reactive sites, and shortened the electron/ion diffusion pathways, resulting in rapid reaction kinetics and enhanced electrochemical performance. The optical and electrochemical properties of the NW–P_2_W_17_La nanocomposite were investigated and compared to those of a densely packed structure. An asymmetric EESD was fabricated using an NW–P_2_W_17_La film as the cathode and a TiO_2_ nanowire film as the anode.

## 2. Results and Discussion

### 2.1. Characterizations of Nanocomposites

Figure 1 depicts the fabrication process of the NW–P_2_W_17_La nanocomposite film. UV/Vis spectroscopy was used to track the growth of the composite film. The deposition of each layer of P_2_W_17_La was accompanied by a linear increase in absorbance, confirming that the growth of the multilayer film obtained via electrostatic self-assembly was uniform and orderly, and that the convenient LbL method can achieve controlled thin-film growth (Appendix A).

For comparison, a pure POM film was fabricated on a fluorine-doped tin oxide (FTO) substrate using the same method (denoted as FTO–P_2_W_17_La). Scanning electron microscopy (SEM) images of the FTO–P_2_W_17_La film and NW–P_2_W_17_La film are shown in Figure 2a,b. The FTO–P_2_W_17_La film exhibits a tightly packed microstructure; the FTO surface is densely covered with aggregates of P_2_W_17_La and polyetherimide (PEI), forming a cobblestone-like morphology. Appendix A shows the vertically grown TiO_2_ nanowires obtained via hydrothermal synthesis. Compared with the pure TiO_2_ NW film, the nanowires in the NW–P_2_W_17_La film are thicker and denser because of the encapsulation of P_2_W_17_La nanoparticles and PEI, forming a 3D sea-cucumber-like microstructure. The TiO_2_ nanowires act as a skeleton on which the POM nanoparticles are deposited, thereby avoiding aggregation and exposing more reactive sites. The cross-sectional image shows that the NW–P_2_W_17_La composite film can reach a thickness of 620 nm, whereas the FTO–P_2_W_17_La film is approximately 110 nm thick. This demonstrated that the 3D sea-cucumber-like microstructure of NW–P_2_W_17_La enhanced the coverage of the surface and provided more electron/ion diffusion pathways. Additionally, as shown in Appendix A, W, P, La, and Ti are homogenously dispersed in the energy-dispersive X-ray spectroscopy (EDS) maps of the NW–P_2_W_17_La composite, confirming that the P_2_W_17_La nanoparticles are uniformly attached to the TiO_2_ nanowires.

The surface topographic features and roughness of the films were examined using atomic force microscopy (AFM) (Figure 2c,d and Appendix A). The thicknesses of the NW–P_2_W_17_La composite film and FTO–P_2_W_17_La measured by AFM were 618 nm and 164.1 nm, respectively, which agree well with the features observed in the SEM images. The root-mean-square roughness of the NW–P_2_W_17_La and FTO–P_2_W_17_La films were calculated to be 108.1 nm and 26.4 nm (with an area of 5 × 5 μm^2^), respectively. The 3D sea-cucumber-like microstructure increases the roughness of the material, thereby increasing the active reaction surface area. 

The transmission electron micrograph (TEM) of unmodified TiO_2_ nanowires scraped from the FTO substrate (Figure 3a) clearly shows the morphology and size of the nanowires. Figure 3b shows the microstructure of the NW–P_2_W_17_La film and the further characterization of the area between the TiO_2_ NW and P_2_W_17_La nanoparticles. After the self-assembly step, the surfaces of the nanowires were coated with P_2_W_17_La nanoparticles (Figure 3b). The TEM-EDS element maps of the NW–P_2_W_17_La film are shown in Figure 3c and Appendix A. The TEM images and corresponding EDS maps confirm that W, P, Ti, and La are uniformly distributed, indicating that the P_2_W_17_La particles enclose the TiO_2_ nanowires to form a core-shell-like structure.

The chemical composition and element valence states of the NW–P_2_W_17_La film were further studied using high-resolution X-ray photoelectron spectroscopy (XPS) (Figure 3d–f). In the deconvoluted XPS spectrum shown in Figure 3d, the strong doublet peaks at 455.8 eV and 461.7 eV are observed, corresponding to Ti 2p3/2 and Ti 2p1/2, respectively, indicating that TiO_2_ was successfully constructed on the FTO substrate [35,36]. As shown in Figure 3e, the strong doublet peaks at 35.1 and 37.2 eV are associated with the binding energies of electrons in the W 4f7/2 and W 4f5/2 orbitals, respectively, indicating that all of the W atoms in the composite film are in the +6 valence state [37]. The W^VI^ atoms can be reduced to the +5 valence state, creating reduced-state POMs. Furthermore, the P 2p peak at 133.0 eV can be ascribed to P_2_W_17_La (Figure 3f).

### 2.2. Electrochromic Performance of Electrodes

The EC characteristics of the NW–P_2_W_17_La film were examined using an electrochemical workstation in conjunction with a UV/Vis spectrometer, and contrasted with those of the FTO–P_2_W_17_La film. Figure 4a shows the visible transmittance spectra (400−800 nm) of the NW–P_2_W_17_La composite film in the bleached and colored states at 0 V and −1.3 V, respectively. Digital photographs of the electrodes show that, when the applied potential is 0 V, the NW–P_2_W_17_La film appears transparent. After applying the −1.3 V negative potential, the film exhibits a deep blue color owing to the reduction of P_2_W_17_La and TiO_2_ (insets, Figure 4a). The maximum optical contrast occurs at 591 nm. The redox reactions and color change of P_2_W_17_La and TiO_2_ are summarized in Equations (1) and (2), respectively [38,39]:[P_2_W_17_O_61_La]^7−^ + x Li^+^ + x e^−^ ⇌ [Li_x_P_2_W_(17−x)_^VI^W_x_^V^O_61_La]^7−^(1)
colorless                                      dark blue
TiO_2_ + x Li^+^ + x e^−^ ⇌ Li_x_TiO_2_(2)
colorless             ligth blue

Under a negative voltage, Li^+^ ions are inserted into the framework of the NW–P_2_W_17_La film, W^VI^ is simultaneously reduced to W^V^, and the film changes color. When a positive voltage is applied, Li^+^ ions are extracted from the film, W^V^ is oxidized to W^VI^, and the film bleaches. TiO_2_ can also act as a cathode EC material and the redox process causes a reversible color transformation from colorless to light blue. But the main role of TiO_2_ in the composite material is to provide a nanowire skeleton base, and its contribution to optical modulation ability is very small. Quantitative EC response analysis was performed using chronoamperometry (CA) and the corresponding in situ transmittance test. Figure 4b shows the in situ transmittance curves of the FTO–P_2_W_17_La and NW–P_2_W_17_La films at 591 nm during subsequent double-potential steps (−1 V and +1 V). The optical contrast of the NW–P_2_W_17_La film (45.6%) is significantly higher than that of the FTO–P_2_W_17_La film (27.2%). The switching times were calculated when the transmittance modulation at 591 nm reached 90%. The NW–P_2_W_17_La composite film exhibits a coloring time of 12 s and a bleaching time of 4 s, whereas the FTO–P_2_W_17_La film requires 10 s and 2 s, respectively. Coloration efficiency (CE) measures the correlation between the optical change and the charge consumption [40]. Figure 4c illustrates the variation in the optical density versus the charge density from the electrolyte to the film. The CE value of the NW–P_2_W_17_La film (91.7 cm^2^ C^−1^) is higher than that of the FTO–P_2_W_17_La film (46.6 cm^2^ C^−1^), indicating that a large transmittance modulation can be achieved with a small charge input. The electrochemical cyclic stability of the NW–P_2_W_17_La film was evaluated over 1000 cycles of double-potential steps at 591 nm. The NW–P_2_W_17_La film retained an optical contrast of approximately 96.9% of its initial value (Figure 4d), whereas the FTO–P_2_W_17_La film retained approximately 82.9% of its initial value (Appendix A). As shown in Appendix A, the morphology of NW–P_2_W_17_La did not change noticeably after 1000 cycles. The experimental results show that the sea-cucumber-like nanostructure conferred improved EC properties compared with the densely packed FTO–P_2_W_17_La structure.

### 2.3. Capacitive Performance of Electrodes

The electrochemical properties of as-prepared films were measured using cyclic voltammetry (CV) and galvanostatic charge/discharge (GCD) tests. The three pairs of redox peaks on the NW–P_2_W_17_La film CV curve are attributed to the redox reaction between W^VI^ and W^V^ and represent typical faradaic behavior during ion insertion/extraction (Figure 5a) [41]. When the scan rate is increased from 10 to 100 mV s^−1^, the shapes of the curves remain largely unchanged, except that the peak potential exhibits a minor shift. The anodic peaks shift toward positive potentials and cathodic peaks shift toward negative potentials, which can be attributed to a reversible but nonideal redox process [42]. To reveal the electrochemical kinetic process, the relationship between the oxidation/reduction peak current and the scan rate was studied. The peak current density i and the corresponding scan rate v follow the mathematical relationship i = av^b^, where a is an adjustable parameter and b is the slope of the linear log i versus log v curve. The magnitude of the slope can be used to characterize the ion storage mechanism during charging and discharging [43]. When b = 0.5, the electrochemical reaction process is governed by the diffusion-controlled process, whereas b = 1 indicates that the process is primarily controlled by the surface reaction, resulting in a capacitance-controlled process. The linear relationship between log i and log v is shown in Figure 5b and the b-values of peaks I and II are 0.68 and 0.60, respectively. This shows that the kinetic process taking place in the NW–P_2_W_17_La film is controlled by ion diffusion and capacitive effects.

Appendix A and Figure 5c show CV curves of the NW–P_2_W_17_La, FTO–P_2_W_17_La, and TiO_2_ NW films obtained at the same scan rate in 1 Mol L^−1^ LiClO_4_/propylene carbonate (PC). The NW–P_2_W_17_La film shows the largest peak current value, demonstrating its high conductivity and low internal resistance characteristics. According to the results presented in Figure 5c, at the scan rate of 20 mV s^−1^, the surface coverage of NW–P_2_W_17_La was determined to be 1.74 × 10^−6^ mol cm^−2^, which is higher than that of the tightly packed FTO–P_2_W_17_La film (3.61 × 10^−7^ mol cm^−2^) [44]. Diffusion coefficients (D) of Li^+^ ions were calculated using the Randles–Sevick equation [45]. The D value obtained for the NW–P_2_W_17_La film is 1.28 × 10^−12^ cm s^−1^, and that computed for the FTO–P_2_W_17_La film is 1.10 × 10^−14^ cm s^−1^. The higher Li-ion diffusion rate obtained for NW–P_2_W_17_La confirms the excellent electrical conductivity of this material.

The GCD measurements of the NW–P_2_W_17_La film are shown in Figure 5d. The shapes of the GCD profiles under different current densities were similar, showing the superior charge/discharge reversibility of the material. Notably, there are three plateaus in the GCD curves and their positions are consistent with those in the CV curves, indicating sound pseudocapacitive behavior. Figure 5e illustrates the volumetric capacitances as a function of the current densities for the FTO–P_2_W_17_La and NW–P_2_W_17_La films. The volumetric capacitance of both films gradually decreases with the increasing current density, which may be due to insufficient ion diffusion during the charge/discharge process at higher current densities. The NW–P_2_W_17_La film achieves a high volumetric capacitance of 190.8 F cm^−3^ at 0.08 mA cm^−2^, whereas that of the FTO–P_2_W_17_La film is significantly lower, reaching 68.9 F cm^−3^ at 0.08 mA cm^−2^. The excellent energy storage performance of the NW–P_2_W_17_La film is primarily due to its 3D sea-cucumber-like microstructure, which is conductive to rapid reaction kinetics.

Figure 5f shows the GCD curve of the NW–P_2_W_17_La film at 0.16 mA cm^−2^, overlaid with the in situ absorbance curve at 591 nm. During the charging process, the NW–P_2_W_17_La film exhibits a noticeable increase in absorbance, whereas during discharging, the absorbance reversibly decreases. There is a clear relationship between absorbance and charge storage; therefore, the storage state of charge can be quantitatively monitored using the absorbance value.

For comparison, some recently reported EC energy storage materials based on POMs and inorganic metal oxides are listed in Appendix A, and our current system exhibits comparable or superior performance. In addition, the transmittance modulation and coloration efficiency of the lacunary and substituted Dawson structures are higher than that of the saturated structure, indicating that the structure and composition of POMs strongly influence their optoelectric properties [46]; therefore, multifunctional properties can be easily adjusted by changing the POM type. In future works, our group will explore the fluorescence regulation ability of the NW–P_2_W_17_La film and POMs with similar structures containing other lanthanides, and develop multifunctional materials based on POMs with multi-band regulation in the visible and fluorescence regions.

### 2.4. Configuration and Performance of EESD

The schematic diagram in Figure 6a shows the configuration of an NW–P_2_W_17_La film electrode-based electrochromic energy storage device. The asymmetric EESD was assembled using an NW–P_2_W_17_La film as the cathode, a TiO_2_ NW film as the anode, and 0.5 M LiI/PC as the electrolyte. TiO_2_ has a high transparency and exhibits only an imperceptible light blue color in the reduced state. In addition, it is widely used in EC and lithium-ion batteries [47] because of its fast insertion/extraction and good stability; therefore, the TiO_2_ NW film was chosen as the anode of the device. Figure 6b shows digital photographs of the EESD under different voltages. The color of the device undergoes a considerable change from transparent to dark blue, which confirms that the EESD exhibits excellent EC properties and can be applied to visualize and quantify the energy storage state. The optical contrast of the EESD reaches 48.6% at 605 nm (Figure 6c), which complies with the requirement for practical applications. To determine the energy state of the device, the change in light transmittance of the EESD was recorded at different voltages. As shown in Appendix A, as the applied potential increases, the EESD gradually changes from transparent to dark blue, and the transmittance value exhibits a clear correspondence to the applied voltage. This shows that the energy storage level of the device can be visualized and quantified. In addition, high switching speeds of 15 s and 4 s are achieved for the coloring and bleaching processes of the EESD, respectively (Figure 6d), and the CE reaches 47.6 cm^2^ C^−1^ (Figure 6e). The cyclic charge/discharge curve exhibits a maximum areal capacitance of 5.72 mF cm^−2^ at 0.15 mA cm^−2^ (Figure 6f,g). Furthermore, the coloring/bleaching process of the EESD is highly synchronized with the charge/discharge process (Figure 6h). Figure 6i shows an LED powered by the EESD. The good performance of the EESD is attributed to the high-performance NW–P_2_W_17_La cathode and the good charge balance and storage capability of the TiO_2_ NW anode in the device.

## 3. Materials and Methods

### 3.1. Chemicals

All chemicals were of analytical grade. Fluorine-doped tin oxide (FTO)-coated glass substrates (<10 ohm/sq) were purchased from Pilkington (Toledo, OH, USA). (3-Aminopropyl)trimethoxysilane (APS) and polyetherimide (PEI) were purchased from Aladdin Bio-Chem Technology Co., Ltd.(Shanghai, China) and used without further treatment. K_7_[La(H_2_O)_x_(α_2_-P_2_W_17_O_61_)] (P_2_W_17_La) was prepared according to the literature [48,49,50]. The structure is shown in Appendix A and characterized using infrared (IR) spectroscopy (Appendix A) and ultraviolet/visible (UV/Vis) absorption spectroscopy (Appendix A).

### 3.2. Fabrication of Composite Films

The TiO_2_ nanowire substrate was synthesized via a hydrothermal synthesis technique according to our previously reported procedure [33]. The multiple-layer film was assembled via the LbL self-assembly method. FTO glass coated with the nanowire substrate was soaked in APS for 12 h, soaked in HCl (pH = 2.0) for 20 min, rinsed with distilled water to remove any HCl attached to the surface, and dried under a nitrogen stream. The nanowire-modified FTO glass was soaked in an aqueous P_2_W_17_La solution (5 × 10^−3^ mol L^−1^) for 7 min, washed with distilled water, and dried under a nitrogen stream. The FTO glass was then immersed in PEI (5 × 10^−3^ mol L^−1^, pH = 4) for 7 min. The POMs and PEI have opposite charges and are alternately deposited on the substrate via electrostatic attraction. This procedure was repeated 30 times to obtain the NW–P_2_W_17_La film comprising 30 layers of P_2_W_17_La. For comparison, a pure P_2_W_17_La film (denoted as FTO–P_2_W_17_La), which comprised 30 layers of P_2_W_17_La, was fabricated on unmodified FTO glass using the same method.

### 3.3. Assembly of Asymmetric EESD

An asymmetric bifunctional EESD was assembled using an NW–P_2_W_17_La film as the cathode, a TiO_2_ NW film as the anode, and 0.5 M LiI in PC as the liquid electrolyte. First, the two electrodes were sandwiched together with 1 mm double-sided tape. The electrolyte was then injected into the space with a syringe to fabricate the prototype electrochromic energy storage device. The effective area of the device was 1 cm^2^.

### 3.4. Materials Characterizations

FTIR spectroscopy was conducted using a GangDong FTIR-650 spectrometer (Tianjin, China) between 400 and 4000 cm^−1^. SEM images were measured on FEI Verious 460 L scanning electron microscope (Hillsboro, OH, USA). AFM images were investigated by Icon Bruker microscope (Ettlingen, Germany). TEM images were measured on a FEI Tecnai G2F20 S-TWIN microscope equipped with an energy-dispersive spectrometer (EDS) (Hillsboro, OH, USA). The TEM sample was prepared by scraping a small amount of the film (0.005 mg) off the FTO substate and dispersing the powder in 1 mL of an ethanol solution by ultrasonication for 40 min. XPS analysis were measured on a Thermo ESCALAB 250 spectrometer (Shanghai, China).

### 3.5. Electrochromic and Energy Storage Performance Measurements

The UV/Vis spectra were obtaining using a PERSEE TU-1901 spectrometer (Beijing, China). The CV and galvanostatic charge/discharge (GCD) measurements were performed using an CHI660B Chenhua electrochemical workstation (Shanghai, China) in a three-electrode configuration, where the as-prepared film served as the working electrode, Ag/AgCl (3 M KCl) as the counter electrode, Pt wire/Pt plate as the reference electrode, and LiClO_4_/PC (1 Mol L^−1^) as the electrolyte. The electrochromic and capacitive properties were studied using in situ TU-T9S PERSEE UV/Vis spectrometry (Beijing, China) with an CHI660B Chenhua electrochemical workstation (Shanghai, China). The EESD was characterized using a two-electrode system.

## 4. Conclusions

We synthesized a bifunctional nanocomposite film comprising POMs and TiO_2_ nanowires and constructed an EESD by assembling the film with a TiO_2_ nanowire-film cathode. Compared with a densely packed structure, the loose 3D sea-cucumber-like microstructure provides a larger active surface area and shortens the electron/ion diffusion pathways, leading to uniform and fast redox reaction kinetics and simultaneously enhancing the EC and electrochemical performance of the composite. The designed EESD exhibits large optical contrast (48.6% at 605 nm), high switching speeds (t_c_ = 15 s, t_b_ = 4 s), and high areal capacitance (5.72 mF cm^−2^ at 0.15 mA cm^−2^). The developed device can power an LED bulb and function as a smart energy storage device where the level of stored energy can be visually and quantitatively monitored using rapid and reversible color variation. This work provides a novel paradigm for POM-based EESDs for optical-electrochemical energy applications.

## Figures and Tables

**Figure 1 molecules-28-02634-f001:**
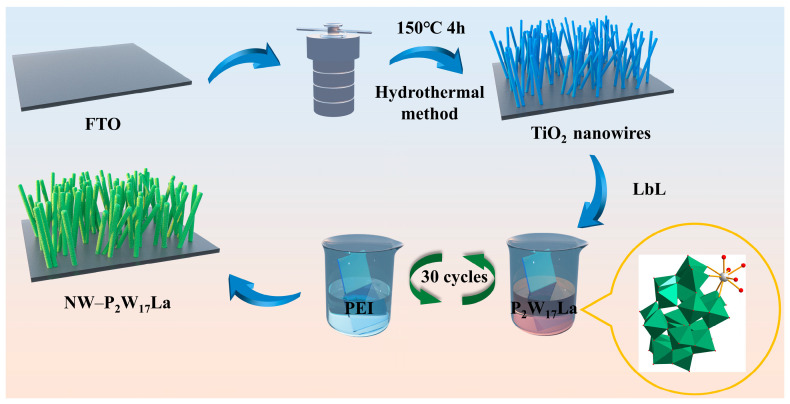
Fabrication process of the NW–P_2_W_17_La nanocomposite film.

**Figure 2 molecules-28-02634-f002:**
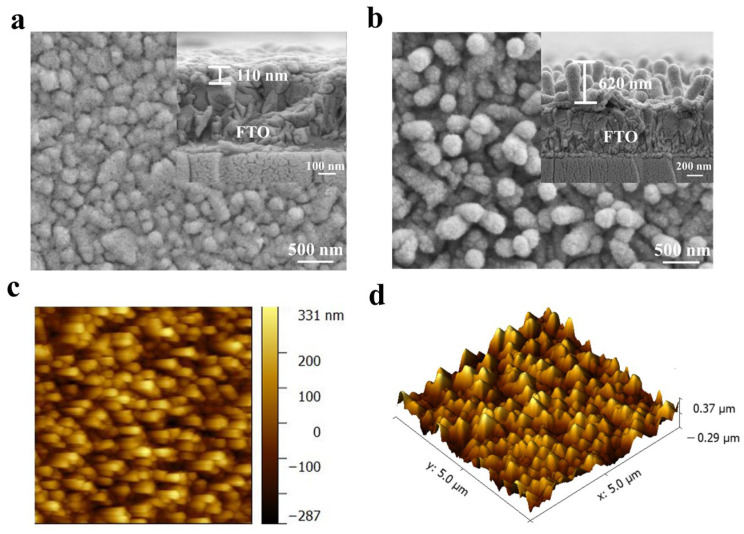
SEM images of (**a**) FTO–P_2_W_17_La and (**b**) NW–P_2_W_17_La composite films (insets: cross-sectional images). (**c**) Two-dimensional (2D) and (**d**) 3D AFM images of the NW–P_2_W_17_La composite film.

**Figure 3 molecules-28-02634-f003:**
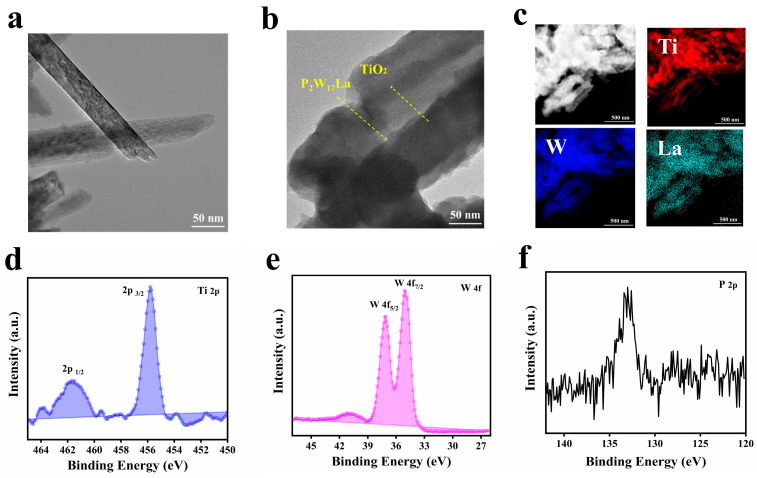
TEM images of (**a**) TiO_2_ NW and (**b**) NW–P_2_W_17_La. (**c**) EDS elemental mapping patterns of Ti, W, and La in the NW–P_2_W_17_La films. (**d**–**f**) Ti, W, and P XPS spectra of the NW–P_2_W_17_La composite film.

**Figure 4 molecules-28-02634-f004:**
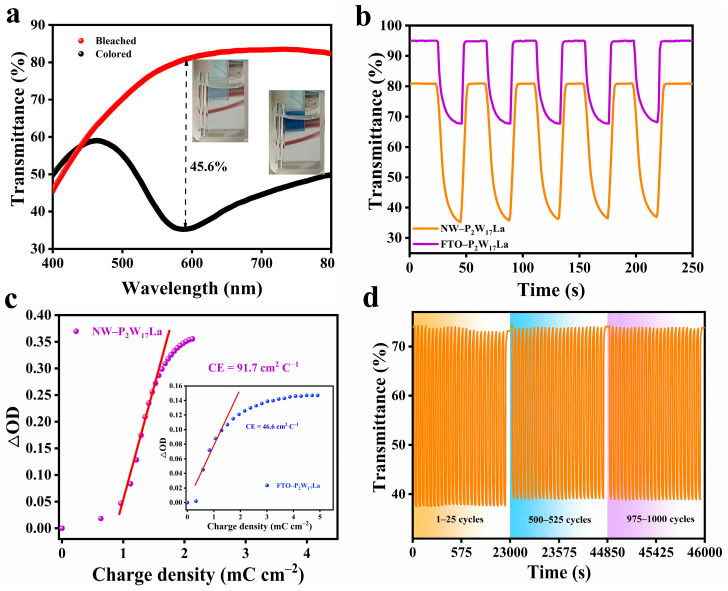
(**a**) Visible transmittance spectra of the NW–P_2_W_17_La composite film in the bleached and colored states at 0 V and −1.3 V (insets: digital photographs). (**b**) In situ transmittance curves of the FTO–P_2_W_17_La and NW–P_2_W_17_La films at 591 nm during pulse voltages (−1 V and +1 V). (**c**) Plot of optical contrast versus charge density for NW–P_2_W_17_La and FTO–P_2_W_17_La films. (**d**) Cycle stability of NW–P_2_W_17_La film at 591 nm under pulse voltages of −1 V and +1 V.

**Figure 5 molecules-28-02634-f005:**
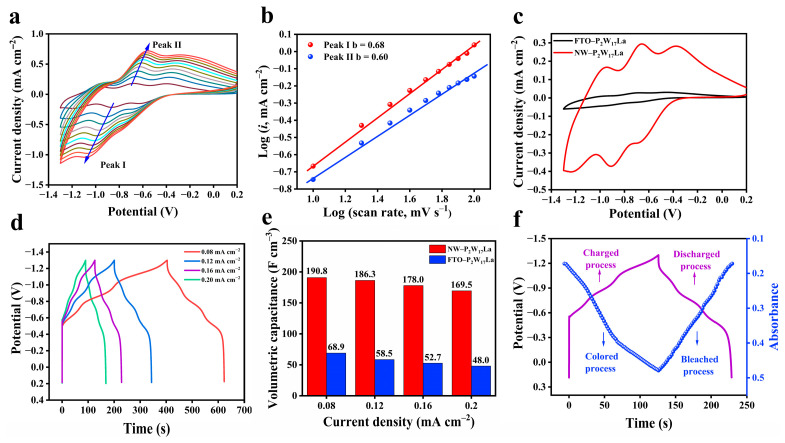
(**a**) CV curves of the NW–P_2_W_17_La film at different scan rates (10–100 mV/s) within the range of −1.3 V to +0.2 V. (**b**) Logarithmic relationship between the oxidation/reduction peak current density (i) and scan rate. (**c**) CV curves for NW–P_2_W_17_La and FTO–P_2_W_17_La films with 20 mV/s. (**d**) GCD profiles of the NW–P_2_W_17_La film at different current densities. (**e**) Comparison of the volumetric capacitances of FTO–P_2_W_17_La and NW–P_2_W_17_La films under various current densities. (**f**) In situ absorbance evolution at 591 nm during the charge/discharge process of the NW–P_2_W_17_La film in the voltage range of −1.3 V to +0.2 V.

**Figure 6 molecules-28-02634-f006:**
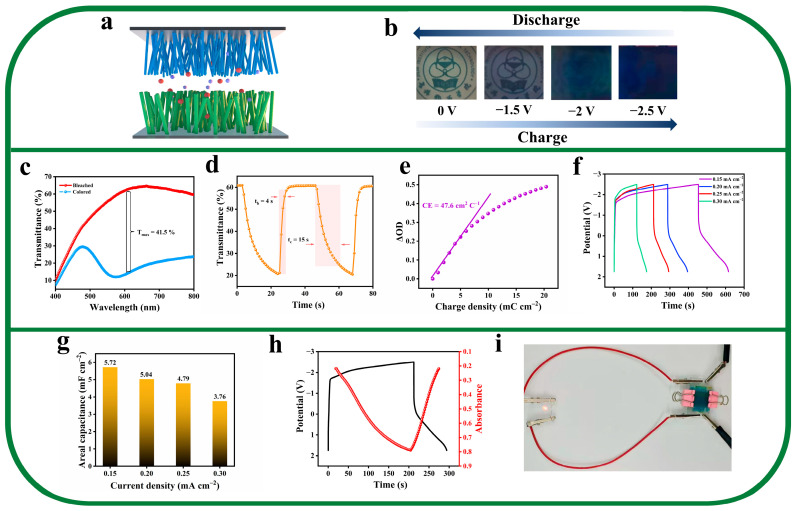
EC and energy storage performance of the EESD. (**a**) Configuration of the asymmetric EESD containing an NW–P_2_W_17_La film. (**b**) Digital photographs of color changes of the EESD under different voltages. (**c**) Transmittance spectra of the EESD in the bleached and colored states at 0 V and −2.5 V in the wavelength range of 400–800 nm. (**d**) Corresponding in situ transmittance responses for 22 s per step measured at 605 nm. (**e**) CE of the EESD at 605 nm. (**f**) EESD constant current charge/discharge curve and (**g**) Area capacitance at current densities from 0.15 to 0.3 mA cm^−2^. (**h**) In situ absorbance evolution at 605 nm during the charge/discharge processes of the EESD. (**i**) Digital photograph of a fully charged bifunctional device powering a red LED.

## Data Availability

Data are contained within the article and Appendix A.

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
