# Peer review of "Sea-Cucumber-like Microstructure Polyoxometalate/TiO2 Nanocomposite Electrode for High-Performance Electrochromic Energy Storage Devices"

_molecules, 2023, doi:10.3390/molecules28062634_

Round 1

Reviewer 1 Report

In this manuscript, the authors synthesized a bifunctional nanocomposite film comprising polyoxometalates(POMs) and TiO2 nanowires and constructed an electrochromic energy storage device by assembling the film with a TiO2 nanowire-film cathode. The conductivity and energy storage properties were also studied. The results of this study are systematic and interesting. It is recommended that this article published in this journal after minor revision.

1. The initial transmittance of NW-P2W17La film is much lower than that of FTO-P2W17La film. What causes it?

2. The microstructure of the composite material described in the paper enhanced the coverage of the surface, exposed more reactive sites, and shortened the electron/ion diffusion pathways. The comparison with the kinetic calculation results of FTO- P2W17La film will be more intuitive and convincing.

3. The author's device uses NW- P2W17La film as the cathode and NW TiO2 as the anode. When the fading of the NW- P2W17La film of the device occurs, the lithium ions are removed from the NW- P2W17La film. In fact, TiO2 is also an excellent electrochromic materia (10.1016/j.joule.2018.12.010; 10.1007/s40820-021-00719-y; 10.34133/2022/9878957). Will the lithium ions enter the NW TiO2 and thus change color, affecting the performance of the device?

Author Response

Response to Referee 1: Comments: Reviewer 1: In this manuscript, the authors synthesized a bifunctional nanocomposite film comprising polyoxometalates(POMs) and TiO2 nanowires and constructed an electrochromic energy storage device by assembling the film with a TiO2 nanowire-film cathode. The conductivity and energy storage properties were also studied. The results of this study are systematic and interesting. It is recommended that this article published in this journal after minor revision. Reply: Thanks for your careful review and valuable suggestions. Your suggestions are very helpful for us to reinforce our manuscript. The initial manuscript has been revised carefully according to your comments. Detailed answers are listed as follows. 1. The initial transmittance of NW-P2W17La film is much lower than that of FTO-P2W17La film. What causes it? Reply: Thank you for the helpful comments. Because we grow the TiO2 nanowire on the FTO conductive substrate, the presence of TiO2 nanowires has a certain effect on the transmittance, resulting in the initial transmittance of NW-P2W17La films being lower than that of FTO-P2W17La. 2. The microstructure of the composite material described in the paper enhanced the coverage of the surface, exposed more reactive sites, and shortened the electron/ion diffusion pathways. The comparison with the kinetic calculation results of FTO- P2W17La film will be more intuitive and convincing. Reply: Thank you very much for this suggestion, According to the results presented in Figure 5c, at the scan rate of 20mV s-1, the surface coverage of NW–P2W17La was determined to be 1.74 × 10−6 mol cm−2, which is higher than that of the tightly packed FTO–P2W17La (3.61 × 10−7 mol cm−2) film. We have calculated the diffusion coefficients (D) of the Li+ ions using the Randles–Sevick equation. The D value for the NW–P2W17La film is 1.28 × 10−12 cm s−1 and that for the FTO–P2W17La film is 1.1 × 10−14 cm s−1. The improved diffusion rate for NW–P2W17La further confirms that the material has excellent electrical conductivity. (Please see line 203-210) 3. The author's device uses NW-P2W17La film as the cathode and NW TiO2 as the anode. When the fading of the NW- P2W17La film of the device occurs, the lithium ions are removed from the NW-P2W17La film. In fact, TiO2 is also an excellent electrochromic material (10.1016/j.joule.2018.12.010; 10.1007/s40820-021-00719-y; 10.34133/2022/9878957). Will the lithium ions enter the NW TiO2 and thus change color, affecting the performance of the device? Reply: Thank you for the helpful comments. The below figure shows the transmittance spectra of TiO2 NW film in the colored and bleached states at wavelength from 400 to 800 nm. When the electrode is cathodically polarized (from 0 to −2V), the film electrode has no discernible color change. It can be seen that the TiO2 NW film shows a very small transmittance modulation of 1.2%. Therefore, even if lithium ions are inserted into TiO2 NW, the overall light transmittance of the device is basically unchanged, and the performance of the device will not be affected. 

Reviewer 2 Report

The manuscript by Yang et al. describes POM/TiO2 nanocomposite electrodes for electrochromic energy storage devices. The research is well-organized, the preparation data are reproducible and characterization gives evidences for proposed composites formation. So, the research meets the following scope of Molecules: Inorganic Chemistry, Physical chemistry, Materials science, Nanoscience and Photochemistry. I have only few issues to be addressed before paper acceptance:

1) What is the role of La in the composite structure? The presence of La is not so cheap from commercial point of view and the second question will be about comparison with pure polyoxotungstates? Is it possible to fabricate the same electrode with plenary K6[P2W18O62] and compare the activities? In my opinion some discussion concerning this should appear in the main text.

2) It should be interesting to compare AFM pictures of initial sample and this sample after 1000 cycles of coloration/bleaching. These data should be added to the main text.

Author Response

Response to Referee 2:

Comments:

Reviewer 2: The manuscript by Yang et al. describes POM/TiO2 nanocomposite electrodes for electrochromic energy storage devices. The research is well-organized, the preparation data are reproducible and characterization gives evidences for proposed composites formation. So, the research meets the following scope of Molecules: Inorganic Chemistry, Physical chemistry, Materials science, Nanoscience and Photochemistry. I have only few issues to be addressed before paper acceptance:

Reply: Thanks for your careful review and valuable suggestions. Your suggestions are very helpful for us to reinforce our manuscript. The initial manuscript has been revised carefully according to your comments. Detailed answers are listed as follows.

  1. What is the role of La in the composite structure? The presence of La is not so cheap from commercial point of view and the second question will be about comparison with pure polyoxotungstates? Is it possible to fabricate the same electrode with plenary K6[P2W18O62] and compare the activities? In my opinion some discussion concerning this should appear in the main text.

Reply: Thank you for the helpful comments. Our group has been committed to the design and fabrication of multifunctional electrode materials based on POMs, and previously designed a series of POM-based EC and dual-function electrochromic-energy storage material. We found that the structure and composition of POMs have a great influence on their electrochemical activity. In general, the lacunary and substituted Dawson structures can show enhanced electrochromic performances[1]. We have reported a nanocomposite film NW–P2W18 based on Dawson-type polyoxometalate K6P2W18O62 (P2W18), which exhibits good EC properties with high transmittance modulation (45.1%) and coloration efficiency (69.0 cm2 C-1)[2]. While in the present work, the nanocomposite film NW–P2W17La showed enhanced electrochromic properties of high transmittance modulation (45.6%) and coloration efficiency (91.7 cm2 C-1). Moreover, in subsequent work, further work will be focused on exploring the fluorescence regulation ability of NW–P2W17La film. We will develop multifunctional composite materials and devices based on POMs with multi-band regulation in the visible and fluorescence regions.

As you suggested, we discuss this in the main text, please see lines 231-241.

"For comparison, some recently reported electrochromic energy storage materials based on POMs and inorganic metal oxides are listed in Table S1, and our current system exhibits comparable or superior performance. It can also be found from the table that the transmittance modulation and coloration efficiency of the lacunary and substituted Dawson structures are higher than that of saturated structure, indicating the structure and composition of POMs have a great influence on their optoelectric properties [46]; therefore, the multifunctional properties could be adjusted easily by changing the type of POMs. In subsequent work, our group will explore the fluorescence regulation ability of NW-P2W17La film and POMs with similar structure containing other lanthanides, and develop multifunctional materials based on POMs with multi-band regulation in the visible and fluorescence regions."

References

[1] S. P. Harmalker, M. A. Leparulo, M. T. Pope, J. Am. Chem. Soc. 1983, 105, 4286–4292.

[2] S. P. Liu, X. S. Qu, Appl. Surf. Sci. 2017, 184,129–195.

  1. It should be interesting to compare AFM pictures of initial sample and this sample after 1000 cycles of coloration/bleaching. These data should be added to the main text.

Reply: Thank you for the helpful comments. The microstructure of NW–P2W17La film after cycling test was characterized by AFM and added to the SI files. As shown in Figure S7, the morphology of NW-P2W17La did not change noticeably even after 1000 cycles. (Please see lines 170-171)
